# Bugs on Drugs: A *Drosophila melanogaster* Gut Model to Study In Vivo Antibiotic Tolerance of *E. coli*

**DOI:** 10.3390/microorganisms10010119

**Published:** 2022-01-07

**Authors:** Bram Van den Bergh

**Affiliations:** 1Department of Entomology, Cornell University, Ithaca, NY 14853, USA; bram.vandenbergh@kuleuven.be; 2Centre for Microbiology, Flemish Institute for Biotechnology, B-3001 Leuven, Belgium; 3Centre of Microbial and Plant Genetics, Department of Molecular and Microbial Systems, KU Leuven, B-3001 Leuven, Belgium

**Keywords:** antibiotic tolerance, persistence, persister cells, in vivo, gut microbiota, *hipA*, *Drosophila melanogaster*, antibiotics

## Abstract

With an antibiotic crisis upon us, we need to boost antibiotic development and improve antibiotics’ efficacy. Crucial is knowing how to efficiently kill bacteria, especially in more complex in vivo conditions. Indeed, many bacteria harbor antibiotic-tolerant persisters, variants that survive exposure to our most potent antibiotics and catalyze resistance development. However, persistence is often only studied in vitro as we lack flexible in vivo models. Here, I explored the potential of using *Drosophila melanogaster* as a model for antimicrobial research, combining methods in *Drosophila* with microbiology techniques: assessing fly development and feeding, generating germ-free or bacteria-associated *Drosophila* and in situ microscopy. Adult flies tolerate antibiotics at high doses, although germ-free larvae show impaired development. Orally presented *E. coli* associates with *Drosophila* and mostly resides in the crop. *E. coli* shows an overall high antibiotic tolerance in vivo potentially resulting from heterogeneity in growth rates. The *hipA7* high-persistence mutant displays an increased antibiotic survival while the expected low persistence of Δ*relA*Δ*spoT* and Δ*rpoS* mutants cannot be confirmed in vivo. In conclusion, a *Drosophila* model for in vivo antibiotic tolerance research shows high potential and offers a flexible system to test findings from in vitro assays in a broader, more complex condition.

## 1. Introduction

Since their first introduction in the 1940s, antibiotics have been the silver bullets of medical practice [1,2]. They serve both as preventative measures in routine and life-saving procedures and as curative agents when suffering from bacterial infections, and they have transformed our healthcare systems and caused a tremendous increase in life expectancy and quality during the 20th century [3,4]. However, in the last decade, antibiotics’ efficacy has been waning, limiting our treatment options [1,4]. In severe cases, infections have even become untreatable [5,6,7]. Resistance, rendering bacteria capable to grow during treatment, causes this antibiotic crisis [8,9]. Previously, the development of new antibiotics kept ahead of the emergence and spread of resistance, an unavoidable consequence of evolution [10]. Lately, however, antibiotic development has slowed down because of a too low return on the high investments, for example from dropouts at the in vitro to in vivo transition in preclinical phases [11,12]. 

Alternatively, we could slow down resistance evolution via alternative treatment schedules and a stricter adherence to regulated use of antibiotics [13,14,15,16,17]. Nevertheless, evolution of resistance seems inevitable, especially since many bacteria display, to some extent, antibiotic tolerance or the capacity to survive lethal treatments, which strongly predisposes bacteria to develop resistance [18,19,20].

Lately, much research focus has been on studying antibiotic-tolerant persister cells. These phenotypic variants within isogenic bacterial populations are much more antibiotic-tolerant compared to their sensitive siblings (i.e., they are killed at a much lower rate) and hamper complete sterilization and prolong treatment [21,22,23]. Recent findings even show that, like resistance, antibiotic tolerance and persistence are under strong adaptive selection when bacteria face recurrent treatments [24,25,26,27,28,29]. Worryingly, the increased tolerance also catalyzes the acquiring of genetic resistance [30,31,32]. Targeting persisters will therefore slow down resistance development and prolong the effectiveness of antibiotics. However, myriad persistence mechanisms to be targeted were identified, for example resolving around toxin–antitoxin systems [33,34,35,36,37,38,39,40], the stringent stress response [41,42], low cellular energy status [43,44,45,46], intracellular protein aggregation [47,48] or acidification [49,50,51]. Currently, it is still unclear how all these different mechanisms fit together: are they each specific to certain conditions or do some of them work together and are all of them relevant in clinical settings?

Persistence insights have been predominantly acquired from in vitro, culture-based assays which offer a great level of control but are also artificially simple in comparison to more complex in vivo conditions. More recent studies started to examine persistence in more real-life conditions. As such, reports did not only verified the presence, relevance and evolution of persistence in vivo [30,52,53,54,55,56,57,58,59,60] but also confirmed the in vivo efficacy of some anti-persister treatments [61,62,63,64] and the relevance of in vitro identified persister genes [36,55,65,66,67]. At the same time, some persistence mechanisms were also found to critically depend on host factors [68,69,70,71] or co-residing microbial species and social interactions [70,72] which additionally shows why application of such models is highly important. Unfortunately, the wide implementation is hampered by the specific nature (e.g., pathogen- and infection-specific) and the high technical, financial and ethical requirements of such in vivo models that often are based on rodent hosts. 

Invertebrate models are used to study host–microbe interactions while circumventing the beforementioned issues associated with, and lowering the use of, vertebrate models [73,74,75]. In addition to *D. melanogaster*, *Caenorhabditis elegans* and *Galleria mellonella*, each with their advantages [73,74,76], are the most popular invertebrate hosts in microbial research of, e.g., virulence and immunology, drug development and the microbiome (for reviews: [77,78,79,80,81]). In a few *C. elegans* and *G. mellonella* studies, microbial persisters were found relevant [82,83] and some antipersister compounds have been validated or screened for [84,85]. To my knowledge, *D. melanogaster* has not been used to study bacterial persisters, which is surprising given (1) its evolutionarily conserved innate immune system, (2) the access to genetic and genomic resources, (3) the wide adoption of *D. melanogaster* in other microbial research and (4) the vast and ever-growing knowledge of the fly gut microbiome and its tractability [76,77,86]. 

Here, I explored the use of a practical, whole-animal model with *Drosophila melanogaster* as a host to study microbial persistence in vivo. Specifically, I assessed the effects of high doses of antibiotics from three different classes on the development, survival and feeding rate of the *Drosophila* host. I included conventional, germ-free and gnotobiotic flies that were orally associated with a variety of *E. coli* strains with different in vitro persistence levels and examined them using microscopy. This way, I examined the association of *E. coli* with the fly and the in vivo antibiotic tolerance when flies received antibiotic treatment.

## 2. Materials and Methods

### 2.1. Cultivation of Drosohila melanogaster and Escherichia coli

*Wolbachia*-free *D. melanogaster* Canton S flies with a conventional gut microbiome (CONV) were reared on an autoclaved yeast–glucose diet (10/10 food) containing 10% D-glucose, 10% inactivated brewer’s yeast and 1.2% agar to which 0.04% phosphoric acid and 0.42% propionic acid were added as preservatives. For germ-free or so-called axenic (AX) and axenic flies that were associated with a single bacterial strain of choice (so-called gnotobiotic flies, see below), the preservative acids were omitted from the 10/10 food. For oviposition plates, a few spoons of frozen, concentrated grape juice were added to a deep purple color. For feeding on liquid food, the fly food was adjusted to 5% D-glucose and 10% yeast extract without any agar (5/10 food). Flies were maintained at 25 °C with a 12 h light/12 h dark cycle and transferred to fresh food every two weeks unless stated otherwise. All fly handling occurred gently by swiftly flipping bottles by tapping or using brushes and tweezers while flies were sedated using CO_2_ sorting pads or via cold treatment (15 min on 0–4 °C cold ice). Flies were homogenized via grinding with plastic pestles or in a FastPrep24 bead homogenizer using lysis matrix D in sterile phosphate-buffered saline (PBS). 

The *E. coli* strains that were used in this study are listed in Table 1. Unless stated otherwise, *E. coli* was grown on lysogeny broth (LB; 1% tryptone, 1% NaCl, 0.5% yeast extract) supplemented with 1.5% agar for solid plates. Liquid cultures were incubated at 30 °C while shaking orbitally at 200 rpm unless stated otherwise. When *E. coli* strains were associated with flies, I always used axenic flies that were devoid of their endogenous microbiome. For confirming the absence of endogenous microbial inhabitants in *Drosophila,* I plated homogenized flies on agar plates of modified De Man, Rogosa and Sharpe medium (mMRS; 1.25% vegetable peptone, 0.75% yeast extract, 2% D-glucose, 0.5% sodium acetate, 0.2% potassium dibasic trihydrate, 0.2% ammonium citrate dibasic, 0.02% magnesium sulfate, 0.005% manganese sulfate hydrate, 1.2% agar). 

Antibiotics were added to the cultures, LB plates, 10/10 or 5/10 food for example when assessing the sensitivity of the development of *D. melanogaster* towards antibiotics or when necessary to select for specific bacterial strains or plasmids: ampicillin (AP) at 100 µg mL^−1^ and kanamycin (KM) at 40 µg mL^−1^.

### 2.2. Generation of Axenic Flies

*Drosophila* was depleted of its microbial interaction partners by dechorionation in bleach as described previously [91,92,93]. Briefly, 300–500 flies were mated 1–3 h before the start of a dark cycle, in a mating chamber containing an oviposition plate topped with a thin lawn of freshly made, inactivated brewer’s yeast paste. The morning after, flies were disposed of and eggs were harvested in a sieve. After this point, all steps were conducted in laminar flow cabinets using aseptic techniques to avoid any contamination while working with AX or gnotobiotic material. Eggs were sterilized 3 times in 0.6% bleach for a total of 5 min and washed 3 times in sterile, deionized water to remove residual bleach. Dechorionated, sterilized eggs were deposited on sterile fly food (±60–100 per Falcon) and allowed to develop to the adult stage. 

During development, the AX state was confirmed in various ways. First, fly food was inspected visually. In case growth of yeast or mold was detected, the vial was disposed of. AX larvae also churn up the 10/10 food differently in comparison with conventional (CONV) larvae. They do not penetrate as deep into the food and preferentially churn at the side of vials. Additionally, the consumed food is darker and more humid than is the case for CONV larvae. Together, this creates a distinct, black layer of consumed food in vials with AX *D. melanogaster.* Vials without such layering were closely monitored. In the pupal stage, vials were checked for their germ-free state by sampling the 10/10 food with a sterile probe (e.g., toothpick, micropipette tip, pestles). Similarly, homogenates of freshly eclosed, female flies were sampled. Both samples were streaked on LB and mMRS agar and incubated at 25 °C (up to 1 week). Only when all parameters were positively evaluated, batches of flies were used and considered AX.

### 2.3. Stability of Antibiotics in 10/10 Fly Food

The stability of AP, amikacin (AM), and ofloxacin (OF) in 10/10 food was assessed for their use in long-term treatments. To this end, I made use of a bioassay that was previously described [87]. The 10/10 agar plates containing antibiotics were incubated at 25 °C and sampled at regular intervals. Discs of 5 mm were cut out using a micropipette tip and deposited on top of a cation-adjusted, Mueller Hinton agar plate that was inoculated with a lawn of the bioreporter strain *Bacillus subtilis* ATCC 6051. Plates were incubated for 24 h at 37 °C and inhibition halos were recorded. Concentrations were deduced from a calibration curve with known concentrations spanning 2-fold dilution from 1024 (AP and AM) or 256 (OF) down to 2 ng/µL. A two-phase association Equation (1) was fitted to these calibration data:*SpanFast* = (*P* − Y_0_) × *PercentFast* × 0.01,*SpanSlow* = (*P* − Y_0_) × (100 − *PercentFast*) × 0.01,(1)Y = Y_0_ + *SpanFast* × (1 − exp(−K_Fast_ × *X*)) + *SpanSlow* × (1 − exp(−K_Slow_ × *X*)),
where *X* is time, Y is the halo diameter that starts at Y_0_ and ascends to Y_0_ + *SpanFast* + *SpanSlow* with two phases, *P* is the plateau value of Y, K_Fast_ and K_Slow_ are rate constants and *PercentFast* is the percent of the signal due to the fast phase. 

### 2.4. Development Dynamics on Antibiotics

Eggs were generated as described in Section 2.2 and were either used directly or made germ-free as described in Section 2.2. Using a brush, ±40 eggs were deposited per vial on sterile 10/10 food with either AP, AM, OF or without antibiotic. Concentrations of antibiotics were 10, 40 or 100 µg mL^−1^ for OF and 10× higher for AP and AM. Afterwards, vials were incubated and checked for their development status daily by recording the number of pupae and eclosed flies. These numbers were then divided by the estimated number of deposited embryos to allow comparison between treatments. Within experiments, 4–5 vials were assigned to each treatment, and the experiment was repeated 4 independent times. 

### 2.5. Feeding Rate in Adult Flies

The feeding rate of AX and CONV flies was recorded using an adjusted capillary feeding (CAFE) assay [94] that was recently introduced in the lab [95]. I provided liquid food to flies via glass, graduated micropipet tubes (Drummon Scientific) with a capacity of 5 microliters. Three of such capillaries were loaded with 5/10 food with/without antibiotics and inserted in the cap of a Falcon tube. Six males, age 2–3 days post-eclosion (dpe), were added to these tubes. Capillaries were recorded daily for decreases in volume of food and replaced with new, fully loaded, capillaries. To correct for evaporation, I included empty controls of which the average loss of volume was subtracted from all the other samples. To limit evaporation, I included a sterile piece of Whatman paper to which 500 µL of sterile deionized water was added at the start in each tube. Additionally, dead flies were recorded daily, and this number was accounted for when expressing feeding rate per fly. Dead flies were assumed to not have fed over the 24 h period before discovery. Within experiments, 4–5 vials were assigned to each treatment and the experiment was repeated 2 independent times. To estimate the antibiotic dosage during treatment of flies, the weight of a male fly is approximated by 500 µg [96] and the human dosage was obtained from the Merck MSD [97].

### 2.6. Generating Gnotobiotic Flies

To generate associations between flies and *E. coli*, a recently described protocol was adopted [98]. Throughout the study, whenever *E. coli* was associated with *Drosophila*, I always made use of germ-free (AX) flies to associate one strain of *E. coli* with at a given time. As the resulting flies have an engineered microbiome, I also call them gnotobiotic flies or gnotobiotics. First, AX adults were generated via the bleach method that was described in Section 2.2. Confirmed AX, male flies (±2–3 dpe; ±60 per tube) were starved for food and water for >4 h at 30 °C to synchronize and maximize the subsequent feeding across flies and samples [98,99,100]. Next, flies were fed for 1 h on fly food on which a bacterial lawn was inoculated with 50 µL of a 1:10, PBS-diluted overnight *E. coli* culture (±5 × 10^6^ colony forming units (CFU) per vial). Afterward, flies were flipped onto sterile 10/10 food and incubated for another six days before the start of the experiments. 

### 2.7. In Vivo Bacterial Load in Absence and Presence of Antibiotic Treatment

To check the effect of antibiotic treatment on the bacterial load in adult male flies, I switched six gnotobiotic flies (i.e., *E. coli*-associated axenic flies) per vial to either solid 10/10 food or to liquid 5/10 food in the CAFE assays as described in Section 2.5 either with or without antibiotics. After the desired treatment time (replacing the capillaries daily and recording feeding rate and survivorship), bacterial loads per fly were estimated by pooling 1–5 living flies (to avoid the known postmortem bacterial proliferation [101]), rinsing them twice with PBS at 4 °C, homogenizing them using a bead homogenizer with lysis matrix D and plating the homogenates using a WASP spiral plater on LB agar (supplemented with antibiotics if necessary). CFUs were determined using a Synbiosis spiral plate counter and normalized to the number of flies that were sampled. 

### 2.8. Microscopy

Microscopy was performed on either an upright fluorescence Leica AF 6000 LX microscope with a DFC365 FX CCD camera or a Zeiss LSM710 confocal setup with a T-PMT detector. Briefly, intact guts were dissected from adult flies using tweezers, needles and a stereoscope. Next, Citifluor AF1 mounting fluid + 2 µg mL^−1^ DAPI was added to the gut sample for staining *Drosophila* cell nuclei and stability purposes. For oil objectives (100× or 63×), a coverslip with a droplet of oil was added on top before imaging. In the mounting fluid, samples could be stored for several days before recording but samples for estimating the growth status using pTimer were recorded on the day itself. On the Leica system, 20× images were recorded using standard fluorescent filter sets. The confocal microscope used lasers at 405, 488 and 561 nm with 410–496, 499–571 and 578–696 nm filters to image DAPI, GFP and dsRed signals, respectively. Exposure times, sensitivities and pinhole diameters were kept as constant as possible across different images and days.

Images were analyzed and transformed either in Zeiss Zen Black software making use of the automated stitching of tiles option or in Fiji [102], the platform that extends the ImageJ2 1.53f51 environment [103], in which the following plugins and core processes were frequently used: *Subtract Background, Stack >> Z Project, Adjust >> Brightness/Contrast, Color >> Channels Tools* and *Split/Merge Channels, Tools >> Scale Bar* and *Stitching* [104]. Additionally, GIMP 2.10.28 was used to fuse images that could not be stitched together in Fiji. Adjustments of brightness and contrast were applied to the entire image. Together with the cropping of images, these manipulations were performed for increasing visual representation only, and always in a highly similar fashion across different images.

### 2.9. Statistics

Statistical analyses were performed in Rstudio (PBC, Boston, MA, USA) or Prism 9.2 (GraphPad Software, San Diego, CA, USA). All visualizations were done using GraphPad Prism 9.2. In Rstudio, I constructed linear models (lm) without random effects using the standard R *stats* package and linear mixed models (lme) using the *lme4* package. For analyzing developmental dynamics, I used Cox mixed models from the *survival* and *coxme* package. For the survival data under capillary feeding, zero-inflated models were constructed using the *pscl* package. For post hoc tests, the *emmeans* package was used with the appropriate multiple testing corrections. For more generic tasks, packages such as *EnvStats, graphics, ggplot2, stats* and *multcomp* were also used.

## 3. Results

### 3.1. Effects of Antibiotics on D. melanogaster

As a first step to using *Drosophila* flies as an in vivo model studying the effect of antibiotic treatments on gut-residing bacteria, I tested the direct effects of antibiotics on the fitness of the *D. melanogaster* host. The included antibiotics represent three major, bactericidal antibiotic classes, i.e., aminoglycosides (amikacin), β-lactams (ampicillin) and fluoroquinolones (ofloxacin). Being approved for and widely used in human medicine, these drugs might not be expected to have large toxic effects. At the same time, the antibiotics could trigger different and/or more extreme responses in *Drosophila* than in humans, especially since I here wanted to apply high drug concentrations, up to 100 µg/mL for ofloxacin (OF) and 1000 µg/mL for amikacin (AM) and ampicillin (AP). High doses should ensure that in vivo drug levels are well above the minimum inhibitory concentration (MIC; at least 100-fold), which effectively induces the bacterial killing in vitro, a prerequisite to study antibiotic tolerance [21]. 

To distinguish between the direct effects of antibiotics on flies themselves and the indirect effects elicited through a diminished or changed microbiota (as reported earlier for mating [105] or feeding preference [106] or for development altogether [107,108]), I included both axenic and conventional groups of *D. melanogaster*.

#### 3.1.1. Antibiotics Induce Strong Mortality in Developing *D. melanogaster* Larvae

To test for effects of antibiotics on *D. melanogaster*, I exploited the well-described developmental dynamics of the fruit fly [107]. Timings of the developmental changes from embryos over pupae to the adult stage were already proven to be highly sensitive towards nutrient quality of the diet [109], and these dynamics therefore should allow picking up subtle effects of antibiotic treatments. Embryos were reared for 14 days on a solid nutrient source containing any of the three antibiotics at three concentrations with a drug-free condition as a control. Under these conditions—where antibiotics were confirmed to remain reasonably stable (Appendix A)—dechorionated, axenic larvae feeding on antibiotics displayed a strongly increased mortality compared to their conventional counterparts or untreated controls (Figure 1A). The average mortality of ±30% across all antibiotics and concentrations in the axenic groups strongly exceeded the mortality of the untreated axenic group and of the treated conventional groups that were statistically insignificant compared to the mortality of the untreated conventional larvae (Figure 1A). Antibiotic-treated, axenic larvae that reached the pupal stage developed successfully into adults and did not suffer any additional lethality during this stage (Appendix A). 

Comparing the developmental dynamics, I detected the frequently reported, slower development of larvae with a diminished microbiome [107,108] as the axenic group was delayed compared to the conventional group without any antibiotics (Appendix A). Likewise, as also expected from their high overall mortality (Figure 1A), the antibiotic-treated, axenic groups displayed a retarded development compared to the treated, conventional groups and the untreated axenic group (Appendix A). However, upon disregarding the mortality in the larval stage, I could not detect any additional development delay in the treated, axenic groups compared to the untreated axenic control, while, globally, an axenic state remained delayed compared to the untreated, conventional group (Appendix A). In other words, antibiotic treatment seems to increase mortality only in axenic groups while, with regards to development dynamics, the greatest delay is caused by an axenic status. Of note, most ampicillin- and amikacin-treated, conventional groups showed a strongly diminished microbiome in the adult stage at the end of the experiment (determined qualitatively by comparing streaks of homogenized flies), resembling that of the axenic state obtained by dechorionation. Agreeing with this, these groups also displayed a small increase in development time (Appendix A). Together, these data suggest that the direct effect of antibiotics in the axenic groups results from an interaction between high antibiotic concentrations and the fragility of the dechorionated embryos. 

#### 3.1.2. Antibiotics Do Not Impact Mortality or Feeding of Adult *D. melanogaster* Flies

I turned to the adult stage to examine direct effects of antibiotic treatments on the fly itself. Indeed, the aforementioned large fitness effect on axenic larvae does not necessarily impede the use of adult *Drosophila* as a gut microbiome model in antibacterial research. To this end, we followed the feeding rate and survivorship of conventional and axenic adults during capillary feeding, with similar treatments as in the development assay, offering a daily recording of daily liquid food (+antibiotic) uptake and mortality (Figure 1B). 

Overall, feeding rates were low on the first day and gradually increased over time (Figure 1B) to reach a daily food uptake of about 1.5 µL fly^−1^. Such a feeding profile potentially implies a form of adaptation in flies as they shift from the spatially abundant, solid nutrient agar to a spatially restricted and liquid-type food and resembles previously reported results [94,95]. No significant differences were found in the feeding patterns of axenic groups compared to their respective conventional counterpart per treatment or between treated groups and their untreated controls (Figure 1B). Therefore, antibiotic-containing food does not change the daily food uptake and food uptake is not different between axenic and conventional flies. 

Fly mortality in the capillary feeding setup, in contrast to the large differences for larval mortality in the development assay, only showed minor, statistically insignificant differences (Appendix A) but also indicated less, instead of more, mortality in the axenic groups. Together, these data confirm that adult *D. melanogaster* flies are receptive for oral, high-dose antibiotic treatments. 

### 3.2. E. coli Associates with Adult D. melanogaster and Locates Preferentially in the Crop

Endogenously, the gut microbiome of *D. melanogaster* contains a range of species, dominated by members of the *Acetobacteraceae, Enterobacteriaceae* and *Lactobacillaceae* families, both in lab as well as in wild *D. melanogaster* [110,111,112,113]. Furthermore, it was previously shown that *D. melanogaster* is a highly permissive host towards an even more expanded set of microbial partners in the lab [92]. Since *E. coli* is one of the most widely studied bacterial species, especially in the context of antibiotic-tolerant persisters [21,114], I here wanted to test whether *D. melanogaster* can act as a host for *E. coli* to allow in vivo assessment of bacterial antibiotic tolerance.

#### 3.2.1. Bacterial Load of *E. coli* Associated with *D. melanogaster* Increases over Time

To avoid any impact of the residing endogenous microbiome on my results, I associated starved, axenic adult males by feeding them briefly with an inoculum of an *E. coli* lab strain, resistant to KM and containing *venus* that encodes yellow fluorescent protein, which was deposited on fly food. Next, I transferred the flies back to sterile solid fly food and incubated them for another week (see Section 2). At this point, flies showed variable bacterial loads, containing an average of ±1500 CFUs with up to ±32,000 CFUs per fly (Figure 2A), which were confirmed to be the applied *E. coli* lab strain via their kanamycin resistance and fluorescence. Associations with *E. coli* do not impact feeding, as gnotobiotic flies show feeding dynamics in the CAFE assay that are highly similar to the previously observed feeding trends of axenic flies (Figure 1B) and to the feeding pattern of a group of axenic flies of the same age (Appendix A).

At the end of the CAFE assay, the bacterial load significantly increased, reaching an average of ±30,000 CFUs per fly (Figure 2A). This increase is independent of the CAFE setup, as flies of the same age, maintained on the regular solid food source, showed a similar increase (Figure 2A). The much lower survivorship of *E. coli*-associated flies in the CAFE setup compared to previous axenic groups in the same assay (Appendix A and Figure 1B), while surprising, should not be considered a confounding effect in the context of the increased bacterial load as care was taken to only sample living flies. 

One could argue that the increase in bacterial load indicates virulence and pathogenicity. However, the used strain is not known to be pathogenic, and axenic flies of the same age show similar increased mortality when maintained in the CAFE setup (Appendix A). Furthermore, *E. coli*-associated or axenic flies of the same age, maintained on solid food, do not show any increased mortality (Appendix A). Thus, not the association with *E. coli*, the CAFE setup nor the age of flies alone can explain the increased mortality. Instead, an interaction between fly age and the CAFE setup seems to lie at the basis of the increased mortality, and altogether, *E. coli* seems to associate with *D. melanogaster* in a quite stable fashion. 

#### 3.2.2. *E. coli* Is Preferentially Present at the Crop of the *Drosophila* Digestive Tract

To validate the presence of *E. coli* in the digestive tract of *D. melanogaster,* I made use of the fact that the *Drosophila* gut can be readily dissected and *E. coli* can be easily fluorescently labeled. The initial Venus tag proved to be too faint for in situ detection; therefore, I made use of a bright green fluorescent tag that is constitutively expressed from a plasmid (see Section 2). As expected, when examining guts of axenic flies directly after the feeding stage during association, bacteria can be found throughout the gut and dispersed rather evenly (Appendix A). After an additional rearing of a week on sterile solid food, *E. coli* cells are still clearly present, at high abundance in the crop and occasionally at other locations in the digestive tract when associated with food boluses (Figure 2B–D and Appendix A). 

When shifted to liquid food during the CAFE assay, *E. coli* remains clearly detectable inside the *Drosophila* gut with a preference for the crop (Appendix A). Furthermore, when applying confocal microscopy at a higher resolution, bacterial cells were identified to regularly occur in clumps and patches inside the crop, indicative of growth of microcolonies or biofilm organization (Appendix A). 

In conclusion, orally presented *E. coli* indeed associates with *D. melanogaster* by residing in its digestive tract and, more specifically, in the crop.

### 3.3. In Vitro Identified Persistence Mutants Show Mixed Results during In Vivo Antibiotic Treatments

With the confirmation that *D. melanogaster* can be treated orally with high doses of antibiotics and that *E. coli* can reside in the gut of the fly, I set out to test the in vivo antibiotic survival of *E. coli* and, more specifically, of in vitro identified persistence mutants to benchmark the herein developed model. To this end, I associated axenic flies with various *E. coli* strains and switched them to the CAFE setup where the antibiotic dose during treatment can be accurately estimated via the food uptake. 

As is evident from the bacterial loads of *D. melanogaster* associated with a wild-type *E. coli* strain (Figure 3A), *E. coli* displays high tolerance in vivo towards antibiotic treatments. Clearly, the high survival after 24 h can be explained by the low food, and thus antibiotic, uptake during this first day in the CAFE setup (Appendix A and as shown previously in Figure 1B). However, at 48 and 72 h, flies feed regularly on antibiotic-containing liquid food (Appendix A), with a food uptake (0.94–1.23 µL day^−1^) that, already for the lowest applied concentrations, results in an antibiotic dosage exceeding the advised dose in humans (Appendix A). Despite such a high dosage, bacteria tolerate the in vivo treatment very well and at least 72 h of treatment is needed to significantly lower bacterial loads. Since ofloxacin resulted in inconsistent lower bacterial loads, I continued using ampicillin and amikacin for at least 72 h. 

Next, I confirmed that the *hipA7 E. coli* mutant, which is widely used in persistence research and results in increased persistence levels in various conditions [21,34], also displays increased survival towards antibiotic treatment inside the *Drosophila* gut (Figure 3B). Both the *hipA7* mutant and the parental wild-type strains show a similar increase in untreated flies. While antibiotic treatment inhibits this increase, the loads of the *hipA7* mutant remain at a higher level after 96 h treatments with both AP and AM. When examining the Δ*rpoS* and Δ*relA*Δ*spoT* mutants, two mutants that are frequently implicated in a lower fraction of tolerant persisters in in vitro studies [21], it becomes apparent that not all in vitro findings can be extrapolated to the more complex in vivo condition of the fly’s gut. Indeed, the Δ*rpoS* mutant does not show a decreased survival upon in vivo treatment. Moreover, while the Δ*relA*Δ*spoT* mutant initially seems impaired in its in vivo antibiotic tolerance, such a conclusion is clearly hampered by the confounding factor that this mutant also associates considerably less successfully with *D. melanogaster* in absence of any antibiotics. 

In an initial attempt to explain the increased tolerance of the *hipA7* mutant in vivo, I made use of a biosensor for growth status as it was shown in vitro that the *hipA7* mutation causes increased persister levels by increased activation of the growth-inhibiting HipA toxin. The plasmid pTimer expresses a dsRed variant that quickly matures to a green fluorescent intermediate fluorophore which then slowly matures to its red fluorescent end product [58]. As such, cells that have not divided for a long time will contain both red and green products, while quickly dividing cells will dilute the red signal and only display green fluorescence. From the confocal images, it is striking that there is quite some diversity in growth rates among individual cells as deduced from the Timer fluorescence, with some green cells that are actively growing and (many) orange cells that show a slower growth (Figure 3D,E). However, while *hipA7* cells seem to display a slower growth rate, in-depth quantitative comparison is far from trivial when doing in situ imaging and working with *z*-stacked images. Therefore, a formal proof of the link between slow growth rate and tolerance is the subject for future experiments. 

Taken together, my data do indicate that some findings on bacterial persistence discovered making use of culture-based in vitro assays can be extrapolated to the more complex conditions of the *Drosophila* gut system while others cannot and that, inherently due to the complexity of an in vivo model, drawing strong conclusions about subtle effects affecting antibiotic tolerance can be much harder. 

## 4. Discussion and Conclusions

Here, I explored the use of *D. melanogaster* as a practical model to study bacterial antibiotic-tolerant persisters in the in vivo and more complex condition that is the fly gut. In building the model, I verified that, while axenic larvae suffer from antibiotics, adult *D. melanogaster* flies are very tolerant towards high-dose antibiotic treatments. In addition, when presenting axenic individuals orally with a bacterial inoculum, *Drosophila* flies can host a substantial amount of *E. coli* cells in their gut, specifically in the crop, at a number that increases over time. Furthermore, I showed that the in vivo conditions induce an overall increased antibiotic tolerance in gut-residing bacteria, as 3–4 days of treatment with high antibiotic doses resulted in only limited decreases of bacterial loads. For the in vitro widely used *hipA7* high-persistence mutant, in vivo antibiotic survival was even higher, but the changed tolerance levels of other in vitro persistence mutants could not be extrapolated to the complex gut environment.

Studying *E. coli* in *D. melanogaster* might seem highly irrelevant. Indeed, microbiome compositions of Drosophila never reported the presence of *E. coli* [111,113]. Furthermore, the physiology of a fly gut is, at points, quite different from that of the human gut, e.g., showing more oxygenation [77,115]. Indeed, a Drosophila-based model will never have the relevance of pathogen-specific infection models with large eukaryotic hosts. In fact, it is not at all possible to “humanize” Drosophila models [77]. Nevertheless, they allow testing, with throughput and power and devoid of ethical issues, the effect of various aspects of environmental complexity and unpredictability on antibacterial tolerance, e.g., the structural and temporal heterogeneity in food and antibiotics or the interaction with host cells. In addition, many aspects of the immune response are conserved between Drosophila and humans, and as *E. coli* and *D. melanogaster* are model organisms, a plethora of knowledge and tools are available for both (e.g., mutant libraries, sensors or defined fly food [116]). Furthermore, *D. melanogaster* is very promiscuous towards its bacterial partners, being a capable host for many nonpathogenic bacteria [117] while sensitive to a wide array of important human pathogens [118,119], making it widely applicable in in vivo research on antibiotic tolerance.

The herein found, specific localization of *E. coli* in the crop nicely agrees with some previous findings on other microbes that more naturally occur in the *Drosophila* gut. The Teixeira group showed that a gut symbiont of *Drosophila, Acetobacter thailandicus,* remains associated in a very stable fashion in the specific niche that is the crop and foregut [108]. In addition, *Lactiplantibacillus plantarum,* another member of the natural microbiome, was also shown to be strongly enriched in the crop and cardia region of the foregut [120]. Even in a *Drosophila*-based model for oral infection with the opportunistic pathogen *Pseudomonas aeruginosa*, bacteria were found specifically in the crop [121]. The crop is an organ in most Diptera insects that stores food (and fulfills a range of other functions) and is diverticulated from the linear digestive tract [122]. As such, the crop can offer bacteria protection from physical clearance by the food flux due to peristaltic movements of the gut. Furthermore, antimicrobial peptides are only to a lesser extent produced in the crop [81,122]. Nevertheless, the crop does contract at times to funnel food into the midgut and produces digestive enzymes. The absence of a mucus layer and the peritrophic matrix that protect the epithelial cells in the midgut and confines bacteria to the gut lumen [86] could aid in a better adherence of bacteria and resilience against evacuation due to crop contractions. Similar to what was found for natural symbionts or opportunistic pathogens, I also observe patch-like clusters of *E. coli* cells that seem to be adhered as microcolonies to the crop epithelial cells [108,120,121], although in-depth research needs to further confirm these findings. 

The bacterial loads that I obtained after association by feeding strongly resemble those that were previously found in a setup where associations were already started in the larval stage [117]. However, compared to the latter setup, if restricting the use of the model to the adult phase, the association-by-feeding strategy offers higher flexibility and additional checkpoints. The obtained numbers furthermore remained stable over time. While I switched flies back to sterile solid food after feeding on the bacterial inoculum, it is hard to speculate what these stable bacterial loads mean in terms of stability of the association between *E. coli* and *D. melanogaster*. I cannot formally exclude that bacteria cycle from food to gut, gut to food and vice versa and that associations are quite unstable. Indeed, it is generally believed that associations, even between naturally occurring microbial gut residents and *D. melanogaster*, are highly flexible and that bacteria need to be replenished orally to maintain stable in vivo densities [123], although two genuinely stable symbionts have been described [108,120]. Very frequent (≥daily) replacement of the sterile food would offer a formal examination in the case of *E. coli* and likely result in declining bacterial numbers over time. However, several of my findings do support some form of stability. To start, a similar bacterial load was observed when flies were maintained in the empty tubes of the CAFE assays where the capillaries with sterile food were replaced daily. While coprophagy could still occur in these vials as the vials themselves were not replaced, it is then striking that antibiotic treatments imposed during CAFE do not result in a stronger decrease (as antibiotics would have excluded further ex vivo proliferation needed to maintain stable bacterial loads). Furthermore, the number of bacteria even increases, which matches with the in vivo detection of microcolony growth patterns and the fact that the Timer growth sensor shows cells that are actively growing. Clearly, a more detailed examination is needed to examine the stability of the association between *E. coli* and the fly.

Bacterial antibiotic tolerance was very high despite the high applied antibiotic dosage. While perhaps surprising at first, antibiotics are known to be less efficient/officious in vivo, although the reasons are not always clear and multiple explanations exist [56]. Here, the heterogeneous in vivo conditions might be an important contributor. Indeed, a heterogeneous in vivo growth status was found among individual bacteria which previously resulted in a delayed eradication of *Salmonella* by antibiotics in mice [58]. A slightly lower growth rate in the more antibiotic-tolerant *hipA7* mutant was apparent from confocal microscopy; however, differences were small and future detailed quantification is needed. In addition, in vivo exposure to stress (e.g., limited nutrient availability, digestive enzymes, immune system, acidity) can further induce antibiotic tolerance [20]. To end, while unlikely given the high dosage and the fact that the increase in bacterial loads was effectively inhibited by treatment, the host and its microbial partners (here single *E. coli* strains) can modulate drugs, leading to lower effective concentrations [124]. The nontransferability of the generally in vitro observed low persistence of the Δ*relA*Δ*spoT* and Δ*rpoS* mutants to the in vivo setting could also be understood in the context of the more complex in vivo conditions. For *rpoS*, even in vitro, some exceptions exist where its deletion in fact can cause an increase in persister levels [125,126], which seems to highly depend on the exact environment and used antibiotic. My results for the fly model further confirm that the environment is extremely important for the precise role of RpoS in antibiotic tolerance. For the Δ*relA*Δ*spoT* mutant, a similar explanatory statement could be made. In addition, such a double mutant lacking ppGpp production is known to be highly crippled and, as shown by my results, clearly has a severely impaired fitness in the fly’s gut even in absence of antibiotics, which acts as a strong confounding effect for any observed differences in in vivo antibiotic survival. 

In the future, the use of this model could be expanded to test some of the many other persistence mutants identified in vitro in *E. coli* [127], examine other physiological traits linked to antibiotic tolerance (e.g., ATP levels) or study antibiotic tolerance in a wide array of bacterial strains or species. Furthermore, to examine the effect of interactions between drugs, host and coresiding bacterial communities on antibiotic tolerance, the Drosophila model could be extremely powerful [124]. Indeed, in vitro, it was already shown that the tolerance of L. plantarum towards rifampin is affected by other important members of the Drosophila gut microbiome, namely Acetobacter species [128]. To end, the model would be extremely suitable to perform in vivo experimental evolution under antibiotic pressure. A growing number of evolution experiments selecting for increased persistence have been performed in monocultures in the lab; however, it is unclear how evolutionary trajectories are dependent on the environmental conditions. A recent evolution experiment with L. plantarum in association with Drosophila shows that the host diet is a dominant factor for evolution, but it remains to be seen whether this is also true when including antibiotics. In conclusion, my results presented here show a high potential for the wide application of a Drosophila-based model to examine antibiotic tolerance in vivo.

## Figures and Tables

**Figure 1 microorganisms-10-00119-f001:**
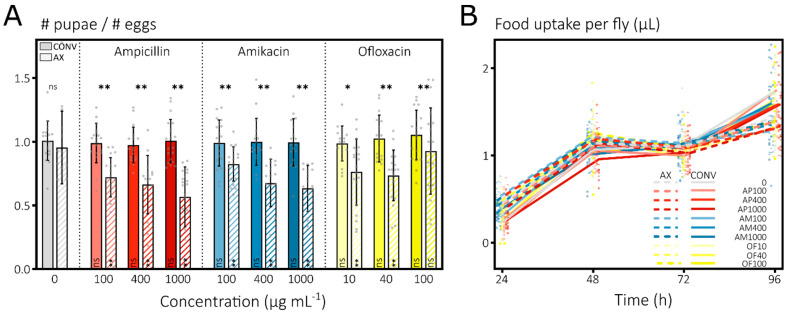
Strong, direct effects of antibiotics on *Drosophila melanogaster* survival and food uptake are limited to the dechorionated, larval stage. (**A**) Antibiotics elicit strong larval mortality in axenic, antibiotic-treated groups. Conventional eggs (CONV, full bars) or dechorionated eggs with eradicated microbiome by bleach treatment (AX, dashed bars) were deposited on the surface of nutrient agar containing either no antibiotic (gray) or ampicillin (red), amikacin (blue) or ofloxacin (yellow) at different concentrations (shades). The ratio of the cumulative number of pupae by day 14 to the estimated number of eggs at the start was plotted (means ± stdevs with *n* = ±800 per group across 20 vials as gray points). Linear mixed model (lme) with post hoc tests for significant differences with the untreated control (Dunnett test per AX and CONV groups, in bar) and between AX-CONV per treatment (Tukey, above bar) (ns, nonsignificant; *, *p* < 0.05; **, *p* < 0.01). (**B**) Antibiotics have no effect on daily food uptake in adult flies. Uptake of antibiotic-free or -containing liquid food was recorded daily for AX (dotted line) and CONV (full line) flies using capillaries (means, *n* = ±60 per group across 10 vials shown as full and half points for CONV and AX). Post hoc tests on an lme with the untreated control (Dunnett) and between AX-CONV per treatment (Tukey) were nonsignificant for all contrasts.

**Figure 2 microorganisms-10-00119-f002:**
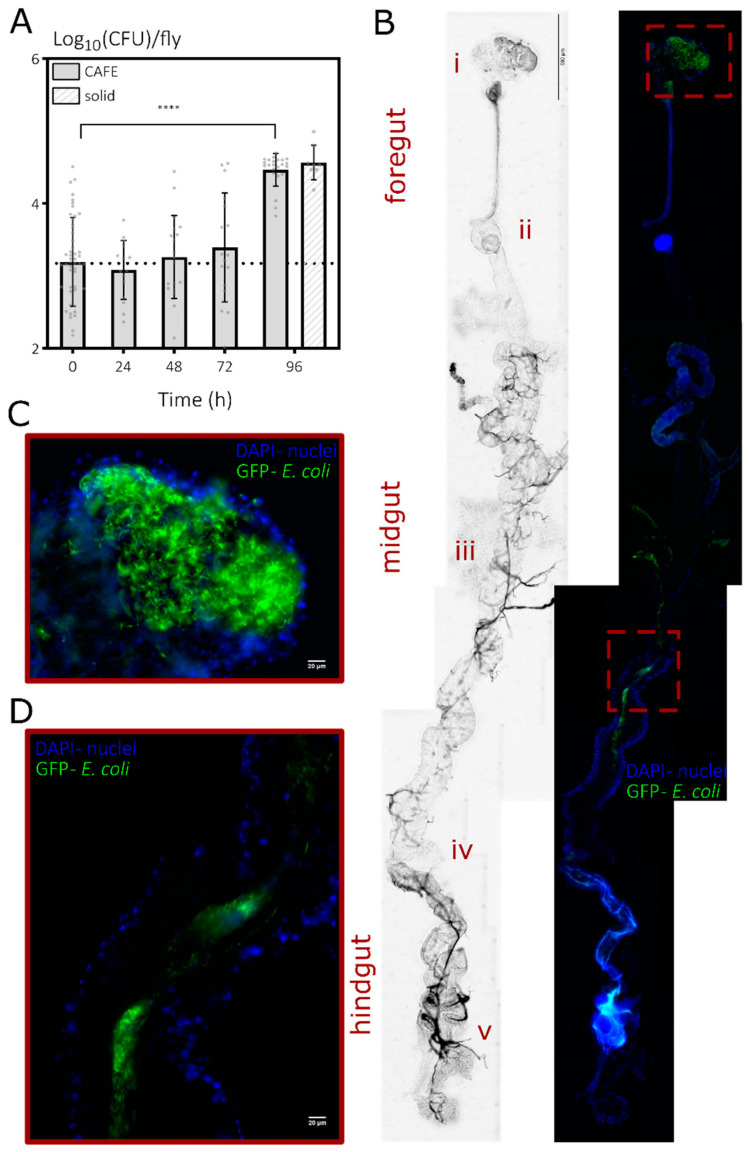
*E. coli* associates with the gut of *D. melanogaster* when presented orally to axenic flies. (**A**) Bacterial load of *E. coli* in the gut of *D. melanogaster* was measured upon transferring gnotobiotic males (9–10 dpe) to a CAFE setup (full bars) or when flies were maintained on solid food (hatched bar at 96 h). Associations were obtained by feeding axenic, starved male flies (2–3 dpe) with an *E. coli* inoculum and a subsequent incubation for another week on sterile solid fly food (see Section 2). Timepoint 0 h is before flies were transferred to the CAFE setup. Bacterial loads are normalized per fly (means ± stdevs with *n* ≥ 8 vials as gray points across ≥2 independent runs) and an lme with post hoc test showed a significant increase to the start at 96 h (Dunnett test above bars with ****, *p* < 0.0001). (**B**) The presence of *E. coli* in the gut of *D. melanogaster* was confirmed using microscopy on dissected guts. Widefield images (left) and a composite (right) of DAPI (staining the *D. melanogaster* nuclei) and GFP (labeling *E. coli* cells) images of one representative gut at 9–10 dpe (a second one is shown in Appendix A). The hallmark structures of the crop (i), cardia (ii), copper cell region (iii), Malpighian tubules (iv) and rectal ampulla (v) are highlighted across the 3 main subdivisions of the fly gut. Whereas *E. coli* cells can be observed in any segment of the gut of gnotobiotic flies, high bacterial densities are specifically found in (**C**) the crop and occasionally in (**D**) other gut regions when associated with the transit of food boluses of which some regions marked with a red square in (B) are magnified. Individual microscopy images were recorded at 20× magnification at a single plane in *z* dimension and adjusted to have an equal brightness when stitched together (see Section 2).

**Figure 3 microorganisms-10-00119-f003:**
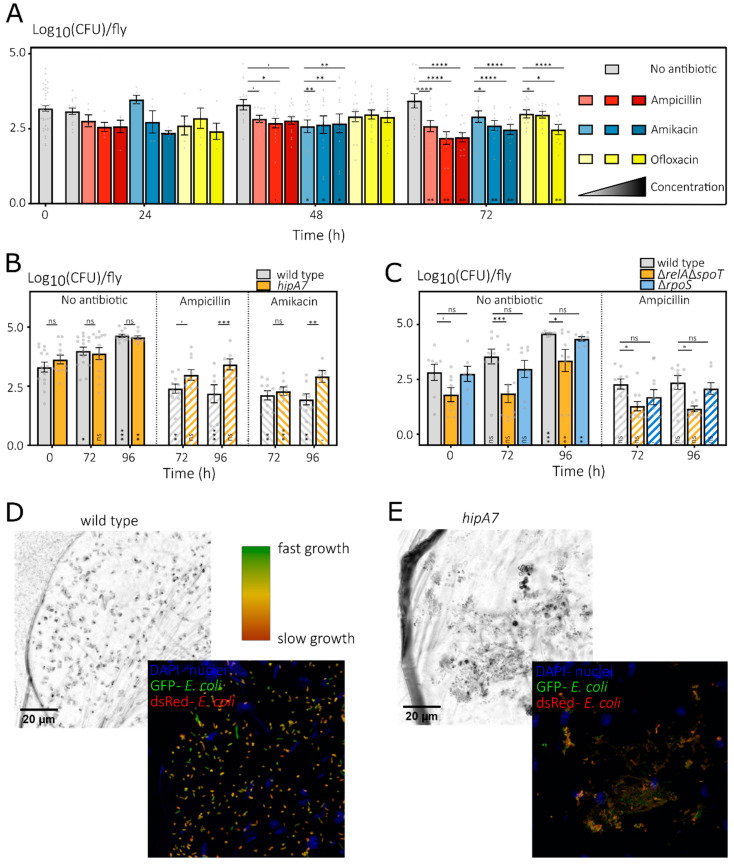
*E. coli* shows high in vivo antibiotic tolerance inside the gut of *D. melanogaster,* with in vitro identified persistence mutants showing mixed results during in vivo antibiotic treatments. (**A**) When treating *D. melanogaster* associated with the SX43 *E. coli* wild-type strain with AP (red), AM (blue) and OF (yellow) at 3 different concentrations (100, 400 and 1000 for AP and AM or 10, 40 and 100 µg mL^−1^ for OF, all in different intensities) via liquid food in the CAFE assay, bacterial loads only start to decrease at 48 h and more consistently at 72 h. Bacterial loads are normalized per fly (means ± SEMs with *n* ≥ 10 vials as gray points across ≥2 independent runs) and an lme with post hoc test shows the significance of the difference compared to the untreated control (gray bars) per time (Dunnett test, above bars) and compared to the untreated start point (Dunnett test, in bars). Nonsignificant comparisons are not annotated as such for visual purposes. (**B**) During treatment with AP and AM (both at 100 µg mL^−1^, hatched bars on the right), the bacterial loads of *D. melanogaster* associated with the *hipA7* mutant (gold) remain higher compared to the loads of those associated with the MG1655 wild-type strain (gray). In contrast, bacterial loads are highly similar in absence of antibiotic treatment (full bars, left). Bacterial loads are normalized per fly (means ± SEMs with *n* ≥ 6 vials as gray points across ≥2 independent runs) and a linear model with post hoc test shows the significance of the difference compared to the untreated start point (Dunnett test, in bars) and between the strains per time and treatment (above bars). (**C**) The in vitro identified low persistence mutants lacking both *relA* and *spoT* (gold) or *rpoS* (blue) do not show a decreased in vivo antibiotic survival when associated with *D. melanogaster* and treated with AP (100 µg mL^−1^). While the Δ*rpoS* mutant is not different from the BW25113 *E. coli* wild-type strain, the loads of Δ*relA*Δ*spoT* are significantly lower during treatment but also in the untreated controls. Bacterial loads are normalized per fly (means ± SEMs with *n* ≥ 8 vials as gray points across 2 independent runs) and an lme with post hoc test shows the significance of the difference compared to the untreated start point (Dunnett test, in bars) and to the wild-type per time per treatment (Dunnett test, above bars). In (**A**–**C**), significance levels are ‘, ns, nonsignificant; *p* < 0.1; *, *p* < 0.05; **, *p* < 0.01; ***, *p* < 0.001; and ****, *p* < 0.0001. (**D**) The MG1655 wild-type strain and (**E**) the *hipA7* mutant show heterogeneous in vivo growth rates as deduced from the Timer sensor. A representative confocal micrograph is shown of a section of the crop of a fly associated with both strains expressing Timer, a dsRed variant that matures quickly to a green fluorescent intermediate and slowly to a red fluorescent end product. The underlying widefield image is shown as a reference, while the fluorescence image is a composite of DAPI (staining the *D. melanogaster* nuclei), GFP and dsRed (labeling *E. coli* cells) images, and a gradient growth scale from green to deep orange was added. The *z* stack of images at 63× magnification was projected onto one plane along with other, image-wide adjustments for visual representation with images of both strains treated similarly (see Section 2).

**Table 1 microorganisms-10-00119-t001:** Bacterial strains used in this study.

**Name**	**Remarks**	**Reference**
*Bacillus subtilis* indicator strain	ATCC 6051	[87]
*hipA7*	MG1655 zde-264::Tn10 *hipA7*, mutant of MG21 with increased persistence	[88]
MG21	MG1655 zde-264::Tn10, parental strain of *hipA7;* also called wild type in figures/text	[88]
*hipA7* pTimer	*hipA7* chemically transformed with pTimer	This study
MG21 pTimer	MG21 chemically transformed with pTimer	This study
SX4	A BW25513 related strain that contains a *tsr-venus* tag in the *lacZ* gene along with a Km^R^ cassette; also called wild type in figures/text	[89]
SX43	SX4 strain where the Km^R^-cassette was removed via expression of FLP recombinase; also called wild type in figures/text	[24]
BW25113	The ancestor of the Keio collection, a derivative of the K12 BD792 strain	[90]
Δ*rpoS*	*E. coli* BW25113 *rpoS::Km^R^* (JW5437-1) that is cured from its Km^R^ cassette	[49,90]
Δ*relA*Δ*spoT*	*E. coli* BW25113 *relA::KmR* (JW2755-1) that is cured from its Km^R^ cassette and in which *spoT* is subsequently deleted	[49,90]
pTimer	pBR322_Timer, expressing *DsRed.T3_S4T* from a constitutive promoter that encodes a green-to-red maturing fluorophore	[58]

## Data Availability

The data presented in this study will be made openly available upon publication in Mendeley Data at doi:10.17632/n6hx7p9trk.1.

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
