# Peer review of "Bugs on Drugs: A Drosophila melanogaster Gut Model to Study In Vivo Antibiotic Tolerance of E. coli"

_microorganisms, 2022, doi:10.3390/microorganisms10010119_

Round 1
Reviewer 1 Report
The manuscript by Bram Van den Bergh (BVB) aims to describe the suitability of Drosophila as a host organism for conducting studies on bacterial persistence. This is an important topic since most studies to date on bacterial persisters were conducted in vitro - as argued by the author. The approach chosen by BVB included the model organism E.coli and variants thereof. BVB established that E.coli can successfully colonise the Drosophila intestine with a preference for the crop as shown by microscopy and enumeration of bacteria by determination of CFUs per fly. Selected antibiotics reduced these CFUs after 72h, albeit not dramatically. A high persister mutant of E.coli (hipA7) resulted in a significant higher number of bacteria surviving Ampicillin/Amikacin treatments. In contrast, mutants that should result in lowered bacterial survival did not produce results that met the expectation.
Major points
1) The major contribution of this work is that it shows that Drosophila can be used for infection studies with E.coli in conjunction with testing the effectiveness of antibiotic treatment. Although an attempt was made to investigate this, little can be learned about persistence in this model system. In my opinion a weakness in the methodological part is that - if I understood correctly - germ-free or axenic fies were only used in the part where the colonisation by E.coli together with a first series of antibiotics testing was performed. All other experiments were done with gnotobiotic flies harbouring the natural microbiota. This could be the reason for unexpected results in case of antibiotic treatments and study on persistence of the strains used. For a better understanding what is going on here I suggest that those experiments be repeated with axenic flies and compared to the results with gnotobiotic flies. Or resolve this issue by considering point #2.
2) The largest part of the results section essentially deals with the establishment of the system (3.1 - 3.2.2). Investigation of persistence and mutants only encompasses one section (3.1) although this area should be the most relevant for the manuscript. One solution could be to completely re-design the paper with only presenting the very solid results 3.1-3.2.2. Then, more work could be invested in the antibiotic tolerance part - where the present findings are preliminary and not comparable to the other data.
Reviewer 2 Report
The manuscript raises a very important problem related to the spread of the phenomenon of antibiotic resistance in the human environment. Many researchers undertake their experiments to identify sources and transmission pathways of antibiotic resistance at the molecular level in order to limit / break this phenomenon. One of the causes of antibiotic resistance of bacteria is their excessive use by humans. Researchers are looking for answers to what factors determine the increased tolerance of bacteria to these chemical compounds, often leading to the acquisition of genetic resistance by them.
The author used Drosophila melanogaster as a model host species to study microbial (E. coli) persistence in vivo. D. melanogaster are the most popular invertebrate hosts in molecular process analysis. Author assessed the effects of high doses of antibiotics from three different classes on the development, survival and feeding rate of the Drosophila host.
The paper is organized and supported with tables and figures in a good organized manner, materials are well illustrated and described, methods of analysis are appropriate and recommended from international organizations.
In conclusion, congratulations to the author of a good manuscript developing research into a Drosophila-based model for in vivo antibiotic tolerance testing.
Round 2
Reviewer 1 Report
Due to the clarifications and changes made by the author the manuscript now is much more consistent - at least in my view. The presented findings will be of interest for researchers trying to find alternatives to rodent models for studying infection/colonisation and the effects of antibiotics in the Drosophila in vivo model system.